# New Insights into the Genetics and Epigenetics of Aging Plasticity

**DOI:** 10.3390/genes14020329

**Published:** 2023-01-27

**Authors:** Jie Zhang, Shixiao Wang, Baohua Liu

**Affiliations:** 1Shenzhen Key Laboratory for Systemic Aging and Intervention (SKL-SAI), School of Basic Medical Sciences, Shenzhen University, Shenzhen 518000, China; 2Guangdong Key Laboratory of Genome Stability and Human Disease Prevention, School of Basic Medical Sciences, Medical School, Lihu Campus, Shenzhen University, Shenzhen 518000, China

**Keywords:** healthy aging, senescence, genetics, epigenetics, epitranscriptome

## Abstract

Biological aging is characterized by irreversible cell cycle blockade, a decreased capacity for tissue regeneration, and an increased risk of age-related diseases and mortality. A variety of genetic and epigenetic factors regulate aging, including the abnormal expression of aging-related genes, increased DNA methylation levels, altered histone modifications, and unbalanced protein translation homeostasis. The epitranscriptome is also closely associated with aging. Aging is regulated by both genetic and epigenetic factors, with significant variability, heterogeneity, and plasticity. Understanding the complex genetic and epigenetic mechanisms of aging will aid the identification of aging-related markers, which may in turn aid the development of effective interventions against this process. This review summarizes the latest research in the field of aging from a genetic and epigenetic perspective. We analyze the relationships between aging-related genes, examine the possibility of reversing the aging process by altering epigenetic age.

## 1. Introduction

Throughout the centuries, there are countless examples of human beings attempting to escape the inevitable: the near-ubiquitous reality of aging and death. While such attempts have been in vain thus far, a number of theories regarding the occurrence of aging have been developed. Some believe that aging is determined primarily by genes [1], while others hypothesize that accumulated cellular damage is the main cause of systemic aging [2]. In fact, aging is a complex process resulting from multiple factors, including genetic and epigenetic molecular markers, such as telomere depletion, genomic instability, and epigenetic alterations [3]. With rapid advances being made in experimental techniques, an increasing body of evidence suggests that these genetic and epigenetic factors are not only individually associated with aging, but that they may work together to drive this process.

Cellular senescence is one of the important factors that trigger aging, and it is also the most widely studied target of aging intervention [4]. Hayflick first proposed the concept of cellular senescence, and he found that mammalian cell cultures divide to a certain stage, then appear senescent or die, which known as the Hayflick limit [5]. Cellular senescence is the process by which cellular functional aging leads to irreversible blockade of the cell cycle, and the two key signaling pathways that control the cell cycle are p53- cyclin-dependent kinase (CDK) inhibitor p21^WAF1/CIP1^-RB and p16^INK4A^–RB pathways [6]. Senescent cells are characterized by stagnation of DNA replication, increased expression of senescence-associated secretory phenotype (SASP), metabolic abnormalities of mitochondria, and lysosomes, changes in the nucleus, resistance to apoptosis, accumulation of DNA damage, epigenetic changes, etc [7]. Studies have found that the use of senolytic (“seno” is senescent, “lytic” meaning destroying) therapy to remove senescent cells can effectively improve aging [8].

Mutations in single genes have been known to significantly impact lifespan since the beginning of the 21st century. For example, Lamin A is one of the main components of the nuclear matrix, and mutations in exon 11 of the LMNA gene damage nuclear structure and function, which manifests as premature aging [9]. Termed progeria, this rare, genetic mutation-linked condition primarily drives aging via epigenetic changes, including altered histone H4 acetylation at lysine 16 (H4K16ac) [10], tri-methylation of H3 lysine 9 (H3H3K9me3) [11], and tri-methylation of lysine 27 on H3 (H3K27me3) [12] and heterochromatin protein 1 (HP1) [13]. Similar epigenetic alterations are also present in Werner syndrome (WS), which is characterized by premature aging and increased susceptibility to cancer [14,15]. Mutations in a single gene can cause drastic changes in lifespan, making us realize that the aging process is not disordered, but dynamically controllable. The mechanism of progeria and normal aging often have many similarities and studying progeria, or aging, can give us insight into the genetic mechanisms of aging and the complex network of factors. This article mainly summarizes the mechanism of aging and the effective intervention methods from the two aspects of the genetics and epigenetics of aging.

## 2. The Genetics of Aging

The lifespans of different biological species lie within a relatively stable range, and there are significant differences between species, which are indicated in the databases (Database of animal ageing and longevity) as summarized in (Table 1). Multiple factors contribute to this difference, including the ratio between body size and heart rate, environmental factors, energy uptake, and genetic factors [16]. In terms of genetic factors, whole genome sequencing has revealed that the mutation rate of non-germline somatic cells between species is an important factor affecting lifespan, with the somatic cell mutation rate having a strong inverse relationship with lifespan and no obvious correlation with body size [17]. It also supports Peto’s paradox, which is the correlation between body size, longevity, and cancer. Interestingly, the study found that not only is the size (number of cells) of individual animals independent of the relative lifespan of the species, but also that large species do not increase the chance of random mutation-induced cancer. [18]. Another important genetic factor is the telomere, which is a repeating double-stranded fragment located at the end of chromosomes in eukaryotic cells, where it maintains the integrity of chromosomes and contributes to controls cell division cycles [19]. Although the relationship between telomere length and species lifespan is somewhat controversial, increasing the length of telomeres in mice has been demonstrated to prolong their lifespan, and telomere shortening rate is an important factor affecting the lifespan of species [20]. The genetic basis of longevity is also closely associated with sex, age, and environmental factors, with the influence of genes on lifespan depending on sex, age, and genetic effects varying between males and females [21]. At present, research concerning genetics and aging mainly revolves around the discovery of gene inactivation and the extension of life expectancy by mutants overexpressing candidate genes. The analysis of large, genome-wide association studies (GWAS) has also resulted in the identification of potential biological markers and targets associated with aging: for example, 27 aging-related gene regions have been found, and a number of these lie close to the gene encoding apolipoprotein E (*APOE*). Recent studies have also demonstrated that APOE protein levels are upregulated in a variety of human stem cell aging models, driving cellular senescence by regulating the stability of heterochromatin [22,23]. The expression of apolipoprotein(a) (*LPA*) and cell adhesion molecule 1 (*VCAM1*) has been found to limit healthy lifespan, LPA performs a role in blood clotting and increases the risk of atherosclerosis. VCAM1 is mainly found on the surface of vascular endothelial cells, and high levels of VCAM1 can also lead to inflammation of blood vessels [23]. Screening genes related to aging through large-scale population data can help improve our understanding of the replication mechanism of aging and provide a good theoretical basis for improving healthy aging and aging-related diseases.

It is particularly noteworthy that sex and aging are closely related. Throughout nature, females generally live longer than males [24]. There are currently two main hypotheses that explain differences in lifespan between sexes: one is sex chromosome differences, and the other is mitochondrial DNA asymmetric inheritance [25]. Sex determination is when male and female sex is determined by different combinations of sex chromosomes [26]. Additionally, many studies have found that sex is profound in terms of longevity. Some aging interventions only work for males and not for females, and vice versa [27]. Between environmental conditions and sex-specific fertility costs and hormones are important causes of gender age differences [24].

**Table 1 genes-14-00329-t001:** Aging-related genetics and epigenetics databases, accessed on 20 December 2022.

GenAge	The ageing gene database	https://genomics.senescence.info/genes/index.html	[28]
AnAge	Database of animal ageing and longevity	https://genomics.senescence.info/species/index.html	[28]
CellAge	Database of Cell Senescence Genes	https://genomics.senescence.info/cells/	[29]
LongevityMap	Human longevity genetic variants	https://genomics.senescence.info/longevity/	[30]
NIA Interventions Testing Program (ITP) Genetics	Conserved longevity gene prioritization	https://www.systems-genetics.org/itp-longevity	[21]
Aging Atlas	Transcriptomics Epigenomics Single-cell Transcriptomics Proteomics Pharmacogenomics Metabolomics	https://ngdc.cncb.ac.cn/aging/index	[31]

Endogenous and exogenous DNA damage can hinder cell function, and DNA repair mechanisms and specific gene mutations are key factors affecting cellular aging [32]. Previous studies have found that genetic mechanisms also underly the unusual longevity of certain groups. For example, some rare mutations carried by centenarians activate genes that inhibit cancer cell metastasis and promote DNA double-strand repair (DDR) [33]. Furthermore, a GWAS study found that *APOE* and G protein coupled receptor 78 (*GPR78*) variants are closely associated with human life expectancy [34]. Whole-exome sequencing (WES) also revealed that rare, longevity-associated coding variants are mainly concentrated in certain pathways of particular relevance to aging. For example, multiple rare variants in the Wnt pathway have been found to counteract the negative effects of APOE4 expression, improving longevity [35]. Aging-related genes and signaling pathways are the core genetic basis of the regulatory network of aging, and the mining of aging-related genes through bioinformatics and experimental exploration of their internal connections will help us unravel the mystery of aging.

### 2.1. Aging-Related Genes and Signaling Pathways

Aging is the most important risk factor for a broad array of diseases, including neurodegenerative diseases, cardiovascular disease, metabolic syndrome, chronic inflammation, and cancer. Additionally, genetic mutations that delay aging have been also found to delay the onset of age-related diseases [36,37]. Genes that regulate aging are relatively conserved among species and are enriched in certain signaling pathways [6] (Figure 1). The association between aging and disease makes fighting aging an even more attractive proposition; it is likely to fight the occurrence and progression of other diseases as well. In this section, we summarize the most important pathways and genes to the aging process.

### 2.2. Nutrient Sensing

Cells rely on nutrient sensing for both the detection of stresses and, ultimately, their survival [38]. Nutrient availability and perception are important material basis for maintaining cell growth and normal function, and cellular metabolic homeostasis imbalance and cellular senescence complement each other [39]. For example, in *Caenorhabditis elegans* (*C. elegans*), mutations in the highly conserved *daf-2* gene, which encodes an insulin-like receptor and regulates the insulin/insulin-like growth factor 1 (IGF-1) pathway, have been found to significantly prolong lifespan [40]. During aging, the mechanistic target of rapamycin (mTOR) signaling pathway is also important for perceiving stress signals and nutrient sensing, and protein translation [41,42]. Genetically inhibiting the insulin/IGF and mTOR pathways has also been demonstrated to extend mouse lifespan [43].

In 1939, researchers discovered that calorie restriction (CR) can ameliorate aging [44]. CR induces various metabolic changes in the body, and crosstalk between CR and proteins related to nutrient sensing-related pathways is an important reason for ameliorate aging [45]. To date, CR has been shown to extend the lifespan of *Saccharomyces cerevisiae, C. elegans*, normal and progeria mouse models, and non-human primate rhesus monkeys; at present, CR represents the most effective lifespan-extending intervention across species [46,47,48,49,50,51,52]. This is due to the fact that most molecular pathways involved in longevity are associated with increased stress resistance [53]. Compared with ad libitum access to food (AL), every-other-day feeding (EOD) increases the healthy lifespan of mice. Dietary restriction has also been found to limit the growth of various types of tumors [54]. The phosphatidylinositol-3-kinase (PI3K) pathway, which is a key insulin signaling component, is an important regulator of CR [50,55]. In addition, restricting the amount of branched-chain amino acids (BCAAs), such as leucine, in the diet has also been demonstrated to prolong the lifespan of *Lmna*^G609G/G609G^ and *Lmna*^–/–^ mice. In terms of physiological aging, a low-BCAA diet reduces weakness, but does not extend lifespan [56]. Thus, achieving CR via the regulation of metabolism and diet represents a promising anti-aging intervention.

### 2.3. Sirtuins

Sirtuins are another gene family that can extend the lifespan of *C. elegans* [57]. They are mainly responsible for regulating cell metabolism, genome stability, gene expression, signal transduction, and important for maintaining the health of the body [58]. There are seven sirtuins in mice and humans, and, under CR, SIRT1 expression is upregulated. This prolongs lifespan and is closely associated with the IGF signaling pathway [59]. Meanwhile, SIRT6 regulates the IGF1 levels and, thus, aging, with SIRT6 overexpression extending lifespan in male mice [27]. Recent studies have also confirmed that SIRT1 is a key protein in the regulation of endothelial cell aging; vascular endothelial cells are essential for maintaining the health and growth of blood vessels. Reducing the expression of SIRT1 in endothelial cells accelerates cellular aging and hinders the normal function of blood vessels [60]. Endothelial cell senescence performs a pivotal role in systemic aging, but the effects can be lessened via the overexpression of SIRT7 [61,62]. Furthermore, SIRT6 expression in endothelial cells has been shown to be important for maintaining heart function [63]. Taken together, these findings indicate that aging-related genes show tissue-dependent effects, and targeting specific types of senescent cells may represent an effective way to treat systemic aging.

### 2.4. Nuclear Skeleton-Associated Proteins

Intranuclear proteins, such as Lamins play an important role in regulating and maintaining the balance of aging and tumors. Mutations in the *LMNA* gene affect aging through a number of mechanisms. For example, Lamin A/C interacts with SIRT1, 6, and 7 and affects their intracellular activity and stability, thereby regulating aging [62,64,65]. Interactions between Lamin A/C and SIRT7 also inhibit the transcriptional activation of long interspersed elements-1 (LINE-1, L1) [66], which stabilizes heterochromatin structure. This inhibits the development of a SASP, such as a type I interferon response that triggers natural immune pathways, and can therefore delay the aging of human stem cells by reducing inflammation [67]. IGF-1/AKT signaling pathway protects cells from apoptosis [68]. Furthermore, recent studies have found that the abnormally processed progerin, which is classically located within the nucleus, is also localized outside it. Here, it interacts with IGF-1R and downregulates its expression, thereby impairing IGF-1/AKT signaling, inhibits cellular energy metabolism and accelerates cell aging [69]. Inhibiting isoprenylcysteine carboxylmethyltransferase (ICMT)-associated activation of AKT-mTOR signaling has been found to improve progeria symptoms [70]. Notably, an *mTOR* hypomorphic allele (*Mtor*^Δ/+^) has also been found to improve aging characteristics and lifespan in *LMNA*^G608G^ mice [71]. Taken together, these findings indicate that Lamin A and nutrient sensing share an intricate, important connection to the aging process.

Furthermore, another protein from the nuclear matrix, Lamin B1, is also closely related to aging. Cells respond to carcinogenic pressure by degrading Lamin B1 through autophagy, thus accelerating cell senescence [72]. Recent studies have found that intranuclear SIRT1 protein is the second major nuclear substrate for LC3-mediated selective autophagy, thus influencing cellular senescence through this degradation mechanism [73].

### 2.5. Immunity and Inflammation

Inflammaging is an important component of aging, which is a pathological phenomenon that brings together our knowledge of age-related chronic diseases, functional decline, and weakness [74]. In the process of aging, the innate and acquired immune system is remodeled, and the reliability and efficiency of the immune system decrease with age, which leads to the upregulation of inflammatory response and the occurrence of related degenerative diseases [75]. The drivers of the inflammatory response mainly include two parts: the degradation of immune receptors/immune sensors and the increase in stimuli that trigger inflammation [36,76]. Inflammation is also the result of lifelong exposure of the immune system to antigenic stimuli and complex genetic, environmental, and age-related mechanisms. Inflammation underlies aging and many age-related chronic diseases, which in turn increases the rate of aging [36]. Excess nutrients are an important factor in inflammation, diet performs an important role in the development and treatment of inflammation and related problems, and CR can slow inflammation and improve aging [77]. The activation of innate immune Toll-Like receptors perform an important role in the aging process, and when Toll-like receptors are knocked out, it can significantly ameliorate the aging of heart-related cells [78]. The Janus kinase/signal transducers and activators of transcription (JAK/STAT) signaling pathway plays an important role in regulating inflammatory response, and the inhibition of JAK/STAT signaling pathway can reduce age-related inflammatory response to a certain extent [79].

Innate immunity plays an important role in the aging process. The cytosolic cyclic GMP–AMP synthase (cGAS)-STING pathway is an important signaling pathway in cells whereby cytoplasmic sensory DNA activates immunity (Figure 1) [80]. During aging, cytoplasmic chromatin fragments (CCFs) leaked from the nucleus, and along with micronuclei or DNA that has escaped from the mitochondria, activate the cGAS-STING pathway, and thus facilitate SASP [81]. SASP promotes the senescence of adjacent or circulatory cells via paracrine signaling [82]. Recent studies have found that yes-associated protein 1 (YAP)/transcriptional coactivator with PDZ-binding motif (TAZ)-mediated control of cGAS-STING signaling is an important molecular mechanism in the regulation of aging in stromal cells and contractile cells. YAP/TAZ is also important for maintaining nuclear envelope stability via the modulation of Lamin B1 expression [83].

### 2.6. Circadian Rhythm

The production and maintenance of circadian rhythm is the result of positive and negative feedback loops regulated by a series of genes associated with the biological clock, including *BMAL*, *CLOCK*, *PER*, *CRY*, *REV-ERB-α, ROR-β*, etc. [84] (Figure 1). The circadian rhythm/clock genes are closely related to aging and two-way adjustment, with aging leading to the transcriptomic reprogramming of circadian genes. For example, the absence of the core clock transcription factor Bmal1 leads to multiple aging-like pathologies in mice [85]. Disturbances in the circadian rhythm accompany the occurrence of aging, and can contribute to the onset and progression of aging-related neurodegenerative diseases [86]. Notably, Salvador Aznar Benitah group and Sassone-Corsi group by comparing mice of different ages, revealed that a low-calorie diet can improve the circadian rhythm of somatic and stem cells, inhibiting the aging process [87,88].

## 3. The Epigenetics of Aging

Recent population studies have found that as aging progresses, genetics have a decreased influence on gene expression. Age-associated epigenetics perform a more important role than genetics in determining which genes in the body are expressed, and this affects susceptibility to disease [89,90]. The relationship between epigenetic modification and age has become increasingly apparent, and the impact of epigenetics on health, lifespan, and longevity has been widely studied. Epigenetic changes that occur during aging may not only serve as indicators of aging, but also drive age-associated transcriptional changes to directly affect the process [91]. Therefore, research thus far has primarily focused on the regulation of gene conditional expression.

### 3.1. Epigenetic Age

Epigenetic processes primarily regulate the aging process by regulated gene expression via dynamic changes in DNA methylation and histone modification, non-coding RNA, and chromatin remodeling [91]. Compared with euchromatin, heterochromatin is more involved in the maintenance of genome stability, which depends on specific heterochromatin-binding proteins, histone modifications, and DNA methylation [92]. Epigenetic changes are closely related to aging, and effect aging process are multifaceted, among which histone modifications and chromatin remodeling respond to epigenetic age from different levels.

#### 3.1.1. Histone Modifications

There are numerous possible histone modifications, with chromatin conformation and gene expression being mainly determined by methylation, acetylation, phosphorylation, and ubiquitination [93]. Genome stability is necessary for maintaining normal physiological functions [94], and alterations to histone modifications occur in specific gene regions during cell aging [3]. During vascular aging, increased levels of histone H3 lysine 4 tri-methylation (H3K4me3) and H3K4 methyltransferase Smyd3 expression in endothelial cells results in the development of SASP [95]. Vascular stiffness increases with age, and in vivo studies in mice have found that H3K27me also significantly decreases during smooth muscle cell aging. H3K27me methyltransferase enhancer of zeste homolog 2 (EZH2) can be used as a new target to improve aging-induced vascular stiffness and fibrosis. [96]. Furthermore, recent studies have found that H3K27me3 is decreased and H3K27me1 is increased during healthy aging of human classical CD14^+^CD16^−^ monocytes [97]. It can be seen that histone modifications in the aging process of different organs are closely related and heterogeneous.

#### 3.1.2. Chromatin Remodeling

Chromatin remodeling and stability are closely related to aging. For example, in damaged mitochondria, acetyl-CoA acts as a signal to induce the accumulation of histone deacetylase complexes (NuRD), thereby mediating chromatin remodeling to regulate body aging [98]. Transcription factor activator protein 1 (AP-1) has also been found to drive SASP by reshaping the accessibility of specific chromatin regions [99]. Furthermore, zinc finger protein with KRAB and SCAN domains 3 (ZKSCAN3) can maintain heterochromatin stability and weaken cellular senescence by interacting with heterochromatin-related proteins [100].

Notably, the relationship between APOE protein and aging has also been revealed recently; APOE affects cellular senescence by regulating the stabilization of heterochromatin in stem cells [22]. HP1α is important for the maintenance of heterochromatin stability. During aging, the loss of HP1α is both an important cause of cellular aging and a biomarker of aging [14]. However, research on non-coding RNAs involved in structural changes in heterochromatin during cellular senescence is still limited, as is our understanding of the underlying mechanisms of action. Non-coding RNA is also involved in maintaining heterochromatin structure, and a recent study found that a combination of *KCNQ1OT1* lncRNA with heterochromatin protein HP1α promoted genome-wide transposon repression. In this manner, the stability of heterochromatin structure was maintained, and cellular aging inhibited [101]. The association between chromatin remodeling and aging is more about homeostasis within the nucleus and the normal expression of genes, and the effects on other aspects of aging are gradually revealed.

### 3.2. Epigenetic Clock

DNA methylation (5′ methylcytosine (5mC)) levels are clearly correlated with age and can be used to predict the chronological age of both blood and certain organs, including the kidneys and liver. Thus, this has been termed the “epigenetic clock” [102]. The epigenetic age of embryonic stem cells is almost zero. It is possible to use algorithms to calculate biological age based on how many sites in an individual’s genome bind to methyl groups (Figure 2). During the passage of mouse embryonic fibroblasts (MEFs) and the physiological aging of various tissues, DNA methylation levels are altered [103]. Radiation and oncogene expression can induce senescence in cells with epigenetic age (EpiAge) close to zero, replicative senescence exhibits increased EpiAge in human cells. Consistent results have also been found in murine cells [103,104]. Notably, fibroblasts taken from patients with Hutchinson Gilford Progeria Syndrome (HGPS) showed a weak correlation with EpiAge [105]. EpiAge has also been found to be closely related to nutrient sensing and the mitochondrial activity [104].

While technical noise is an important limiting factor in the reliability of aging biomarkers and the epigenetic clock [106], the reversibility of the epigenetic clock has become a favored topic of conversation in the field of aging research. Epigenetic reprogramming and the use of extracellular vesicles are two potential methods of reversing the epigenetic clock (Figure 2) and will be discussed below.

#### 3.2.1. Epigenetic Reprogramming

Epigenetic remodeling is an important driver of aging [107]. Dialing back the epigenetic clock through epigenetic reprogramming seems to represent an effective way to reverse the aging process. For example, the partial reprogramming of Yamanaka factor (*Oct4*, *Sox2*, *Klf4*, and *c-Myc* (OSKM)) in progeroid mice has been shown to improve metabolic dysfunction and aging characteristics [108]. However, aging and reprogramming have a more complex relationship, and OSKM can also facilitate the reprogramming process by inducing cell damage and bringing cells into a state of senescent, organizational environment conducive to OSKM-driven reprogramming in neighboring cells [109]. Using adeno-associated virus (AAV) to introduce three transcription factors, *Oct4*, *Sox2*, and *Klf4* (OSK) in aging mice, age-associated visual impairments were successfully reversed [110]. Recent studies have found that the in vivo reprogramming of normal aging mice restores youthful epigenetic characteristics in senescent cells, and significantly reduces the expression of inflammation/aging-related genes [111]. In addition, recent cocktail therapies utilizing reprogramming and senolytic strategies have successfully extended the lifespan of mice [112].

The results of these studies suggest that epigenetic reprogramming represents a highly promising and effective aging intervention. Compared with the introduction of genes to induce reprogramming, compound-induced reprogramming can effectively avoid safety problems associated with traditional transgenic operations [113]. Recently, human somatic cells have been successfully induced into pluripotent stem cells using only small chemical molecules, without relying on exogenous genes [114]. The application of compound-induced cellular reprogramming in aging is yet to be fully investigated, and may become an effective alternative to aging cocktail therapy in the future.

#### 3.2.2. Extracellular Vesicles

Extracellular vehicles (EVs) mainly involved in the removal of excess or unnecessary substances from cells, the maintenance of homeostasis in the intracellular environment, and to serve as messengers during cell-to-cell communication [115]. Using EVs to deliver extracellular nicotinamide phosphoribosyltransferase (eNAMPT) to elderly mice has been found to alleviate age-related tissue function, and to significantly extend lifespan [116]. Recent studies have also found that injecting old mice with adipose mesenchymal stem cell extracellular vesicles (ADSC-sEVs) from young mice improves a number of aging problems in old mice; even reducing tissue epigenetic age [117]. As effective carriers of aging intervention mediator, such as proteins, nucleic acids, and lipids, exosomes have broad application prospects in the future, and will likely be of great research significance.

### 3.3. Epitranscriptomics

Epitranscriptomics focuses on the effects of RNA post-transcriptional modifications on the regulation of gene expression and has provided novel strategies to analyze the epigenetics of aging (Figure 3). RNAs can be covalently modified in a number of ways, including N6-methyladenosine (m^6^A), 5-methylcytidine (m^5^C), N1-methyladenosine (m^1^A), N4-acetylcytidine (ac4C), and N 7-methylguanosine (m^7^G), etc. [118]. Manipulating RNA modifications may represent a useful molecular mechanism through which longevity and stress tolerance might be improved, and we discuss some potential targets below.

#### 3.3.1. m^6^A

RNA m^6^A modification is dynamically regulated by methyltransferase-like (METTL) 3/14, fat mass and obesity-associated protein (FTO)/alkB homologue 5 (ALKBH5), and reader proteins, such as YTH domain-containing family (YTHDF)1/2/3 are involved in the functional outcome. m^6^A is involved in multiple aspects of RNA posttranscriptional processing, including RNA stability, translation, splicing, and export to the nucleus. A number of studies have highlighted the intrinsic link between RNA-modified enzymes and aging [119,120,121], but the relationship between m^6^A readers and aging needs to be studied in greater depth. YTHDF1/2, YTHDC1/2, heterogeneous nuclear ribonucleoprotein A2B1 (HNRNPA2B1), and heterogeneous nuclear ribonucleoprotein C (HNRNPC) protein expression levels have been shown to decrease during replicative senescence [122]. The loss of cell proliferation ability is a significant feature of cell senescence, and Forkhead box M1 (FOXM1) is a key proliferation-related transcription factor that regulates the expression G2/M phase transition genes. Notably, YTHDF1 regulates the translation of FOXM1 in an m^6^A-dependent manner [123], and FOXM1 overexpression has been shown to significantly improve physiological and pathological aging [124]. As aforementioned, Insulin/IGF-1 transmembrane receptor (IGFR) is particularly important for aging, and YTHDF3 expression patterns have been found to be consistent with insulin receptor (INSR) expression during aging [125]. YTHDF3 binds to m^1^A modifications on *IGF1R* mRNA to promote its degradation, thereby inhibiting IGF1R expression [126]. YTHDF3 also regulates the expression of longevity gene forkhead box O-3 (FOXO3) in both m^6^A-dependent and non-dependent ways, thereby regulating immune activation and autophagy [127,128].

#### 3.3.2. ac^4^C

N-Acetyltransferase 10 (NAT10) is the key enzyme involved in the ac^4^C RNA acetylation modification and is essential for normal cell function [129]. Remodelin, an inhibitor of NAT10, improves the nuclear morphology of HGPS [130], and *Nat10* heterozygosity has been shown to extend the lifespan of progeroid mice [131]. Recent studies have found that the ac4C modification can be removed by SIRT7 in vitro [132]. However, the dynamic changes of ac^4^C in vivo and its impact on aging remain to be fully characterized.

#### 3.3.3. m^1^A and m^5^C

Other RNA modifications have also been found to be closely related to aging. Demethylase ALKBH3 of *N*1-methyladenosine (m^1^A) increases during aging and regulates hematopoietic system function [133]. The m^5^C modification is catalyzed by methyltransferase NSUN2/TRDMT1 and oxidized by dioxygenase ten-eleven translocation (TET) to form hm^5^C. NSUN2 can add the m^5^C modification to the p16 mRNA strand, which can inhibit the degradation of p16 mRNA and increase its stability [134].

## 4. Interventions for Aging

The rapid progress in the study of the molecular mechanism of aging has made us realize that the speed of aging and the length of life expectancy can be intervened by humans. To understand the genetic and epigenetic biological mechanisms behind aging, gain insight into the plasticity of aging, and, ultimately, achieve effective intervention in aging [135]. At present, the most effective aging intervention method is still CR, and the development of age-related genetic research has made us realize that aging-related pathways can also be used as targets for aging intervention, such as insulin-like signaling pathways, AMP-activated protein kinase (AMPK) signaling pathways, sirtuins, nicotinamide adenine dinucleotide (NAD^+^), circadian clock, chronic inflammation, cellular senescence, etc [136]. Increasing NAD^+^ levels can significantly ameliorate aging [137]. SASP is an important feature of cellular senescence, SASP can promote the aging of neighboring cells through autocrine or paracrine signaling pathways, and the accumulation of senescent cells is one of the important causes of aging with age. The intervention methods of senescent cells mainly include senolytics that eliminate senescent cells and senomophics that reduce SASP, Senomorphic compounds target pathological SASP signals, and senolytic eliminates potential senescent cells that release harmful SASP factors [138]. The use of senolytics to remove senescent cells and reduce the secretion of SASP can effectively delay tissue or overall aging. The use of drugs or other ways to reduce the expression of SASP can also significantly delay the aging of cells [8] Rapid development of biological theory and technology, gene therapy anti-aging has been widely studied and recognized. Gene therapy that delivers anti-aging genes through AAV introduction has great potential in treating multiple age-related diseases at once [139]. With the innovation of technology, gene editing and aging vaccines will also likely move towards anti-aging applications in the near future [140]. The cocktail of aging therapy combines different anti-aging methods to greatly enrich the anti-aging strategy [112].

## 5. Conclusions

The study of genetic and epigenetic characteristics of aging is essential to the development of targeted treatments against aging. Still, our understanding of the genetics and epigenetics of aging is just the tip of the iceberg. Since the aging process is dynamic and complex, the use of more genomics and epitranscriptomics methods can effectively realize the real-time, dynamic, and multi-dimensional monitoring of the aging process. The genetic and epigenetic changes of aging were combined to select suitable target interventions. Then, a cocktail of therapy through a combination of multiple ways is possible to achieve a reversal of aging. With advances in scientific theories and technologies, personalized anti-aging treatments, improved health during aging, and increased longevity may not be far away. The US Food and Drug Administration (FDA) recently approved the use of the farnesyltransferase inhibitor Zokinvy (lonafarnib) to reduce and prevent the accumulation of defective presenile proteins, thereby reducing the risk of death caused by HGPS. This represents an important step from basic research to clinical application [141]. The Lifespan.io website (Lifespan.io is a nonprofit advocacy organization and news outlet covering aging and rejuvenation research) (Figure 4) provides us with current clinical application information for anti-aging medications or therapies. Notably, the Targeting Aging with Metformin (TAME) research plan involves the study of the anti-aging effects of metformin in a series of nationwide, six-year clinical trials at 14 leading research institutions, involving 3000 non-diabetic patients (65–79 years old) [142]. There is no shortage of exciting bonus scenes in aging research, in which a fancy study used deep neural networks to assess cellular senescence and found that nuclear morphology is an important biomarker of cellular senescence [143]. With the analysis of multiple big data related to the genetics and epigenetics of aging, the combination of artificial intelligence and aging research will greatly broaden people’s understanding of aging. It seems likely that in the near future, even more aging-related treatment modalities will be developed for clinical application. It is of great significance to understand the plasticity of aging process, the plasticity of aging biomarkers and drivers, and the plasticity of aging interventions from the perspectives of genetics and epigenetics, which is of great significance for improving healthy aging.

## Figures and Tables

**Figure 1 genes-14-00329-f001:**
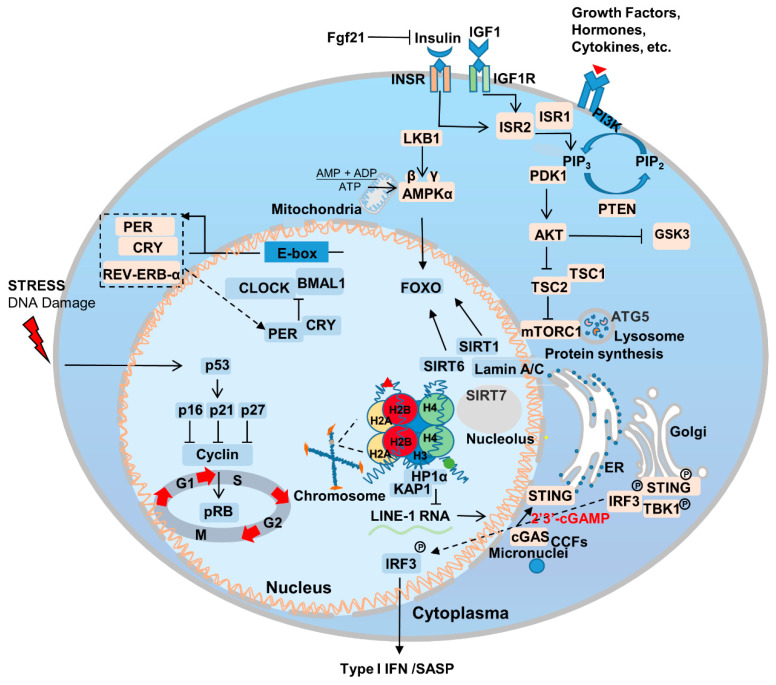
Genetic and signaling mechanisms underlying aging. Aging involves multiple genetic alterations in a range of pathways, including, but not limited to, nutrient sensing, sirtuins, nuclear skeleton proteins, immunity, inflammation and circadian rhythm. PI3K/AKT, AMPK, and mTORC1 serve as the core members of lipid, glucose, and amino acid sensing. Lamin A/C interacts with SIRT1, 6 and 7 to regulate chromatin and intracellular homeostasis. cGAS-STING responds to internal and external nuclear pressures and regulates senescence-associated secretory phenotype (SASP). The feedback regulation of circadian rhythm-associated genes is also affected by other aging-related genes. p16, cyclin-dependent kinase inhibitor 2A; p21, cyclin-dependent kinase inhibitor 1A; p27, cyclin-dependent kinase inhibitor 1B; Rb, retinoblastoma protein.

**Figure 2 genes-14-00329-f002:**
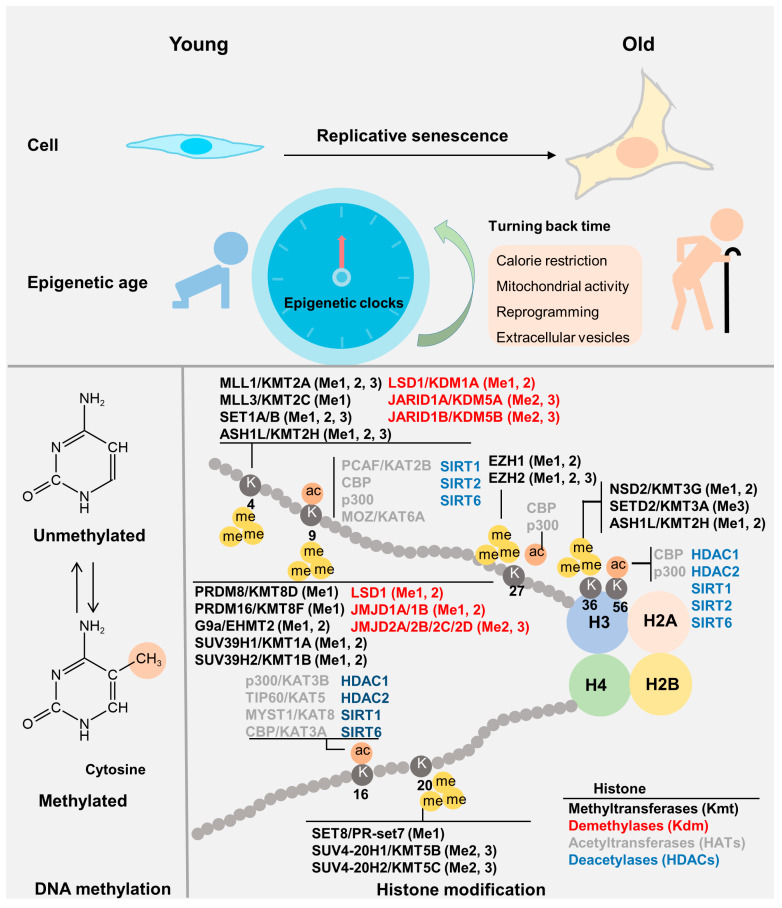
Epigenetics of aging. The “epigenetic clock”, a biomedical measure of aging, primarily includes DNA methylation and histone modification status. During both cellular and systemic aging, epigenetic age increases. The epigenetic clock can be reversed by calorie restriction, mitochondrial activation, reprogramming, and extracellular vesicles. The section on histone modifications partially references the pathway and diagram of cell signaling technology (CST), and mainly lists histone modifications related to aging and regulatory kinases related to histone modifications. Ac, histone acetylation; me, histone methylation.

**Figure 3 genes-14-00329-f003:**
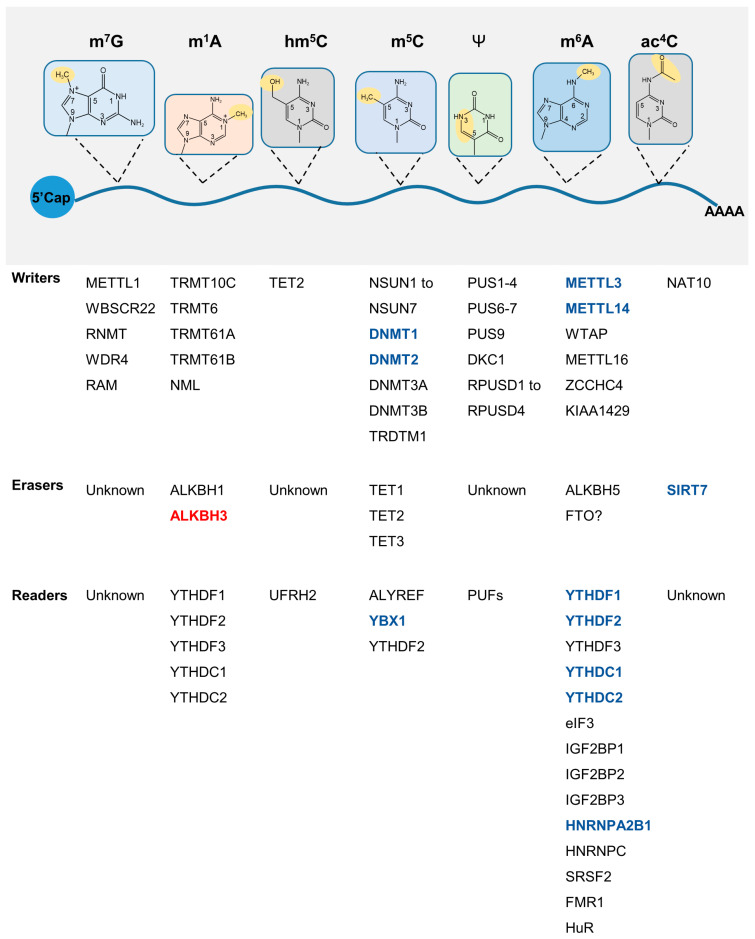
RNA modification and senescence. Seven types of RNA modifications have been studied thus far, including m^7^G, m^1^A, m^6^A, m^5^C hm^5^C, ac^4^C, and Ψ. A summary of the writer, eraser, and reader proteins of the related modifications is presented. Aging-related changes: red represents upregulation, while blue represents downregulation. M^7^G, 7-methylguanosine; m^1^A, *N*1-methyladenosine; m^5^C, 5-methylcytosine; hm^5^C, 5-hydroxymethylcytidine; m^6^A, *N*6-methyladenosine; Ψ, uridine-to-pseudouridine.

**Figure 4 genes-14-00329-f004:**
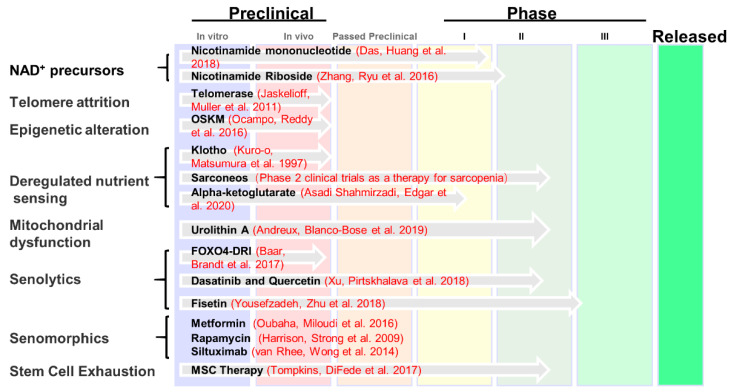
Research and development of clinical anti-aging medication. The information presented in the figure is summarized at https://www.lifespan.io/road-maps/the-rejuvenation-roadmap/, accessed on 20 December 2022. A number of aging-related drugs are currently subject to preliminary exploratory investigations, while other disease-related or anti-cancer drugs have been gradually introduced into anti-aging research. At present, the most promising treatments include drugs targeting NAD^+^, telomere attrition, epigenetic alteration, deregulated nutrient sensing, mitochondrial dysfunction, senolytics, senomorphics, and stem cell exhaustion [60,108,144,145,146,147,148,149,150,151,152,153,154,155].

## Data Availability

Not applicable.

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
