# Peer review of "New Insights into the Genetics and Epigenetics of Aging Plasticity"

_genes, 2023, doi:10.3390/genes14020329_

Round 1
Reviewer 1 Report
Zhang and collaborators made an extensively work putting together the new insights of epigenetics and genetics in terms of aging. The general idea is interesting, and the audience would find it important to the field. However, there are some comments that would contribute to strength and add solid bases to the proposed perspective.
1.- The authors are highlighting the novelty of this work in terms of the aging plasticity. I graphical abstract intersecting the general ideas such as pathways involved/epigenetics modifications/therapeutic options would be found appreciated it by the general audience.
2.- The authors recapitulate and synthesize the relation and epigenetics in aging extensively, but it is not clear for this reviewer where does the previous reviews covered this idea, and where is the novelty contribution? . The authors can, for instance, strength the difference between females and males, or discuss further about it (in case that there are not current works that validate a significant difference)
3.- A deeper discussion about the future perspectives of what would be needed in the filed would add strength to the closing conclusion.
Reviewer 2 Report
The manuscript by Zhang et al. provides an overview of the up-to-date discoveries in the research of aging and anti-aging strategies on both cellular and organismal levels. The authors presented the related research from 3 major aspects with very good clarity: genetics, epigenetics, and epitranscriptome. For the genetics of aging, the authors provided a general overview of aging-related genes and signaling pathways, with certain focus on nutrient sensing, sirtuins, nuclear skeleton-associated proteins, and other conventional aspects such as the circadian rhythm; while touched upon the relatively more recently developed/popular topics of immunity and inflammation. This information is mostly well-established and widely known, yet the authors’ overview could serve as a good base of knowledge for readers who need a general, comprehensive, and essential introduction of aging from the genetics point of view. The authors then addressed with great effort on the topic of the epigenetics of aging, providing an effective summarization of the knowledge in epigenetic aging and epigenetic clock. It is especially valuable that the authors introduced and well-explained the epitranscriptomic aspect of aging-related studies and current discoveries, which is a relatively new and under-studied field of aging research and can be enlightening to readers seeking for new research directions or gaining novel insights in the understanding of the mechanism of aging.
There are certain flaws of this manuscript (which will be commented in detail below). For instance, the authors need to add introduction to certain key concepts in aging before using them as important arguments in the review. Overall, I think this work serves the readers well with the purpose of introducing the essential knowledge in the aging field and should be recommended, given if they address the major and minor points below:
Major points:
1. The authors heavily used the concept of cellular senescence and its related characteristics such as the senescence-associated secretory phenotype (SASP) and the use of senolytics, yet without providing a general and basic introduction to this concept. Though cellular senescence is a well-established concept, for the reader of this article, it will highly improve the efficiency of delivery the message of this review if the authors could add a section to introduce the basics of cellular senescence, such as its discovery (This will also benefit the understanding of replicative senescence as the authors mentioned in the text.), its basic characteristics (such as SASP as well as histone γ-H2AX, which will further help the readers’ understanding of the chromatin structure of aging cells), and the recently rather popular field of senolytics, etc. The concept of cellular senescence will also help the readers understand the perspective of cellular vs. organismal/physiological aging.
2. The main content of the paper addresses well-known concepts and mechanisms from the aging research field. To increase the portion of more novel discoveries and concepts, it will be helpful if the authors structurally emphasize the most recent updates in aging research. Also, the authors touched upon the field of interventions methods towards aging/senolytics and a few anti-aging strategies embedded in the introduction of certain aging mechanism or in the very last paragraph of “conclusion”. In my opinion, it might help to improve the value and structure of this review if there is a separated section of anti-aging strategies or treatment added in the article.
3. Due to the importance and relatively rapid progresses in the field of immunity and inflammation in aging studies, it is suggested to elaborate on the “2.1.3 Immunity and inflammation” section.
Minor points:
1. The title includes the phrase “aging plasticity”, yet the concept of “plasticity” is not emphasized in the text of this article and the meaning of “plasticity” here is a bit unclear -- Does it mean reversal of aging/anti-aging potential, or the multiple levels of aging regulatory mechanism? The authors should either explain the concept of “plasticity” and stress it in the text as a major point or consider changing the use of this phrase in the title.
2. It will be helpful if the author could add more details and in-depth comments/reports of contextual information for some of the reference of related studies so that the readers could have a better understanding of the conclusion made from these studies (e.g. in which species the gene/signaling pathway/intervention was studied; If the study was done in vivo or in vitro only).
3. Some of the sentences in this manuscript are a bit unclear of its meaning or confusing in its message to deliver. Examples are:
-- Line 53-53: The authors should state clearly that the information about lifespans of different biological species are indicated in the databases as summarized in Table 1, not that Table 1 is directly presenting the lifespan/species information.
-- Line 61-63: The meaning of this sentence is not clear. It might help to break this sentence into more sentences to convey a clearer message (and explain why the concept of cancer comes in here).
-- Line 65-66: The statement is too absolute. It might be helpful to say that “(telomere) contributes to the controlling of cell cycles” instead of saying that it “controls cell cycles” as if telomere is the only determining factor in cell cycle.
-- Line 225: The “(selective)” here is confusing. It needs further clarification or adjustment.
-- Line 242-244: Should break into 2 sentences. Also, the phrase “theoretical guidance” is confusing. It needs further clarification.
-- Line 277-279: The meaning of the first half of the sentence is unclear, or the phrase “in cells” should be added before “with epigenetic age”.
-- Line 300-303 The meaning of the sentence is unclear, or the word “senescence” should be changed into “senescent”.
4. There are certain English phrasing and grammar flaws in this article.
For example:
-- Line 80-84: This sentence needs to be separated into two or more sentences.
-- Line 214: Grammar error. No subject in the sentence.
-- Line 238: The word “a” should not be used.
5. Some of the fonts and word sizes need further editing. For example:
-- Line 99-102: Increase of word size for no reason.
-- Second-tier titles (e.g. Line 103) are bigger than first-tier titles (e.g. Line 52) throughout the manuscript.
6. Reference is needed for the sentence in Line 271-273.
7. The sentence in Line 308-310 is a bit too general. If preferred, elaborated information could be added to the suggested section of “anti-aging/senolytic interventions”. The same suggestion applies for Line 314-316.
8. Exosome was mentioned in Line 320 and Line 329. The word exosome cannot be used interchangeably with the phrase “extracellular vesicles (EVs)”, since the former is just a sub-category of the latter. Since the authors are focusing on the general discoveries related to EVs, the word “exosome” does not need to be mentioned in order to avoid confusion.
